# Memory-oriented Decoder for Light Field Salient Object Detection

**Miao Zhang**[*]    **Jingjing Li**[*]    **Wei Ji**[*]    **Yongri Piao**[†]    **Huchuan Lu**

Dalian University of Technology, China

miaozhang@dlut.edu.cn, {lijingjing, jiwei521}@mail.dlut.edu.cn,
{yrpiao, lhchuan}@dlut.edu.cn

## Abstract

Light field data have been demonstrated in favor of many tasks in computer vision, but existing works about light field saliency detection still rely on hand-crafted features. In this paper, we present a deep-learning-based method where a novel memory-oriented decoder is tailored for light field saliency detection. Our goal is to deeply explore and comprehensively exploit internal correlation of focal slices for accurate prediction by designing feature fusion and integration mechanisms. The success of our method is demonstrated by achieving the state of the art on three datasets. We present this problem in a way that is accessible to members of the community and provide a large-scale light field dataset that facilitates comparisons across algorithms. The code and dataset are made publicly available at https://github.com/OIPLab-DUT/MoLF.

## 1 Introduction

Salient object detection (SOD) is the ability to identify the most visually distinctive objects despite substantial appearance similarity in a scene. This fundamental task has attracted lots of interest due to its importance in various applications, such as visual tracking [20, 47], object recognition [43, 10], image segmentation [33], image retrieval [44], and robot navigation [9].

Existing methods can be categorized into 2D (RGB), 3D (RGB-D) and 4D (light field) saliency detection based on the input data types. 2D methods [15, 23, 8, 18, 21, 36, 27, 63] have achieved great success and long been dominant in the field of saliency detection. However, 2D saliency detection methods may suffer from false positives when it comes to challenging scenes shown in Fig. 1. The reasons are twofold: First, traditional 2D methods underlie many prior knowledges in which violations highly pose a risk under complex scenes; Second, 2D deep-learning-based methods are subject to the features extracted from limited RGB data not containing as much special information from RGB-D data or light field data. 3D saliency detection has also attracted a lot of attention because depth maps providing scene layout can improve the saliency accuracy to some extent. However, mediocre-quality depth maps heavily jeopardize the accuracy of saliency detection.

The light field provides images of the scene from an array of viewpoints which spread over the extent of the lens aperture. These different views can be used to produce a stack of focal slices, containing abundant spatial parallax information as well as accurate depth information about the objects in the scene. Furthermore, focusness is one of the strongest information, allowing a human observer to instantly understand the order in which objects are arranged along the depth in a scene [24, 59, 29]. Light field data have been demonstrated in favor of many applications in computer vision, such as depth estimation [16, 48, 64], super resolution [67, 55], and material recognition [51]. Due to the

---

[*]denotes equal contributions.

[†]Prof.Piao is the corresponding author.

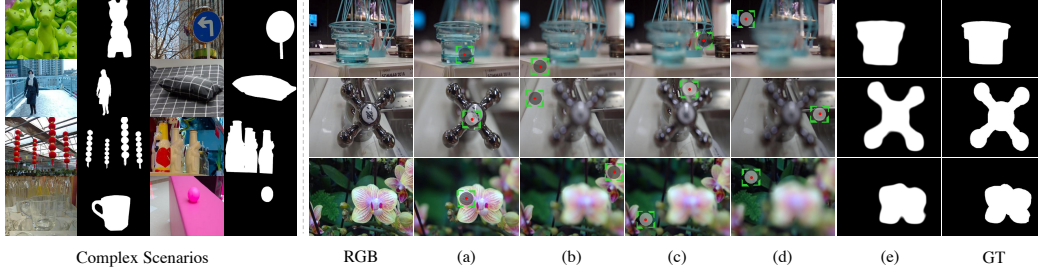

Figure 1: **Left:** some challenging scenes, e.g., similar foreground and background, complex background, transparent objects, and low intensity environment. **Right:** the light field data. (a)-(d) are four focal slices that focus at different depth levels. The green box with red dot represents different focus positions. From our observation, they are beneficial for efficient foreground and background separation. (e) shows our model's saliency results. 'GT' means ground truths.

unique property of light field, it has shown promising prospects in saliency detection [24, 28, 58, 56, 59, 29]. However, deep-learning-based light field methods have been missing from contemporary studies in saliency detection. We have strong reasons to believe introducing the CNN framework for light field saliency detection is an important aspect, as do 2D and 3D methods in SOD.

In order to incorporate the CNN framework and light field for accurate SOD, there are three key issues needed to be considered. First, how do we solve the deficiency of training data? Second, how do we effectively and properly fuse light field features generated from different focal slices? Third, how do we comprehensively integrate multi-level features?

In this paper, we leverage the ideas from light field to confront these challenges. To better adapt our network to fuse features from focal slices, we may neither want to ignore more contribution of the corresponding focal slices where the salient object happens to be in focus, nor destroy the spatial correlation between different focal slices. Therefore, we propose a novel memory-oriented spatial fusion module (Mo-SFM) to resemble the memory mechanism of how human fuse information to understand a scene by going through all pieces of information and emphasizing the most relevant ones. On the other hand, integration of fused features is used for higher cognitive processing. Therefore, we propose a sophisticated multi-level integration mechanism in a top-down manner where high-level features are used to guide low-level feature selection, namely memory-oriented feature integration module (Mo-FIM). The previous information referred to as memory is used in our channel attention to update the current light field feature, so that important and unnecessary features can be distinguishable. In summary, our main contributions are as follows:

- We introduce a large-scale light filed saliency dataset with 1462 samples, each of which contains an all-focus image, a focal stack with 12 focal slices, a depth map, and a corresponding ground truth, genuinely hoping that this could pave the way for light field SOD and enable more advanced research and development.
- We propose a novel memory-oriented decoder tailored for light field SOD. Feature fusion mechanism in Mo-SFM and feature integration mechanism in Mo-FIM enable more accurate prediction. This work is, *to the best of our knowledge*, the first exploitation of using the unique focal slices in light field data for deep-learning-based saliency detection.
- Extensive experiments on three light field datasets show that our method achieves consistently superior performance over 25 state-of-the-art 2D, 3D and 4D approaches.

## 2 Related Work

**Salient Object Detection.** Early works [23, 8, 18, 19, 40, 68, 32, 30, 41, 49] for saliency detection mainly rely on hand-crafted features and prior knowledges, such as color-contrast and background prior. Recently, with the utilization of CNNs, 2D SOD has achieved appealing performance. Li *et al.* [27] adopt a CNN to extract multi-scale features to predict saliency for each super-pixel. Wang *et al.* [50] propose two CNNs to integrate local super-pixel estimation and global search for SOD. Zhao *et al.* [63] utilize two independent CNNs to extract both global and local contexts. Lee *et al.* [26] combine low-level distant map with high-level semantic features of deep CNNs for SOD. These

methods achieve better performance but suffer from time-consuming computation and injure the spatial information of the input images. Afterwords, Liu and Han [35] first generate a coarse saliency map and then refine its details step by step. Hou *et al.* [21] introduce short connections into multiple side-outputs based on HED [54] architecture. Zhang *et al.* [60] integrate multi-level features in multiple resolutions and combine them for accurate prediction. Luo *et al.* [37] propose a simplified CNN to combine both local and global information and design a loss to penalize boundary errors. Zhang *et al.* [62] and Liu *et al.* [36] introduce attention mechanism to guide feature integration. Deng *et al.* [11] design a residual refinement block to learn the complementary saliency information of the intermediate prediction. Li *et al.* [31] transfer contour knowledge to saliency detection without using any manual saliency masks. Detailed surveys about 2D SOD can be found in [3, 2, 4, 52].

In 3D SOD, depth images with affluent spatial information can act as complementary cues for saliency detection [38, 39, 14, 25, 42, 5]. Peng *et al.* [39] regard the depth data as one channel of input and feed it into a multi-stage saliency detection model. Ju *et al.* [25] and Feng *et al.* [14] present saliency methods based on anisotropic center-surround difference or local background enclosure. Zhu *et al.* [66] propose a center-dark channel prior for RGB-D SOD. Qu *et al.* [42] use hand-crafted features to train a CNN and achieve better performance than tradition methods. In [17, 7], two-stream models are used to process the RGB image and depth map separately and cross-modal features are combined to jointly make prediction. Due to limited training sets, they are trained in a stage-wise manner. Chen *et al.* [5] design a progressive fusion network to fuse cross-modal multi-level features to predict saliency maps. Chen *et al.* [6] propose a three-stream network to extract RGB-D features and use attention mechanism to adaptively select complement. Zhu *et al.* [65] use large-scale RGB datasets to pre-train a prior model and employ depth-induced features to enhance the network.

Previous works in light field SOD have shown promising prospects, especially for some complex scenarios. Li *et al.* [29, 28] report a saliency detection approach on the light field data and propose the first light field saliency dataset-LFSD. Zhang *et al.* [58] propose saliency method based on depth contrast and focusness-based background priors, and show the effectiveness and superiority of light field properties. Li *et al.* [56] introduce a weighted sparse coding structure for handling heterogenous types of input data. Zhang *et al.* [59] integrate multiple visual cues from light field images to detect salient regions. However, deep-learning-based light field methods are still in the infancy, and many issues have yet to be explored.

# 3   Light Field Dataset

To remedy the data deficiency problem, we introduce a large-scale light field saliency dataset with 1462 selected high-quality samples captured by Lytro Illum camera. We decode the light field format file using Lytro Desktop. Each light field consists of an all-focus image, a focal stack with 12 focal slices focusing at different depths, a depth image, and a corresponding manually labeled ground truth. The focal stack resembles human perception using eyes, i.e., the eyes can dynamically refocus at different focal slices to determine saliency [29]. Fig. 1 shows samples of light fields in our proposed dataset. From our observation, they are beneficial for efficient foreground and background separation. During annotation, three volunteers are asked to draw a rectangle to the most attractive objects. Then, we collect 1462 samples by choosing the images with consensus. We manually label the salient objects from the all-focus image using a commonly used segmentation tool. By supplying the easy-to-understand dataset, we hope to promote the research and make the SOD problem more accessible to those familiar with this field. The proposed light field saliency dataset provides the unique focal slices that can be used to support the training needs of deep neural networks.

This dataset consists of 900 indoor and 562 outdoor scenes captured in the surrounding environments of our daily life, e.g., offices, supermarkets, campuses, streets and so on. Besides, this dataset contains many challenging scenes as shown in Fig. 1, e.g., similar foreground and background(108), complex background(31), transparent objects(28), multiple objects(95), and low-intensity environments(9).

# 4   The Proposed Network

## 4.1   The Overall Architecture

We adopt the widely utilized VGG-19 net [46] as the backbone architecture, drop the last pooling layer and fully connected layers, and reserve five convolutional blocks to better fit for our task, as

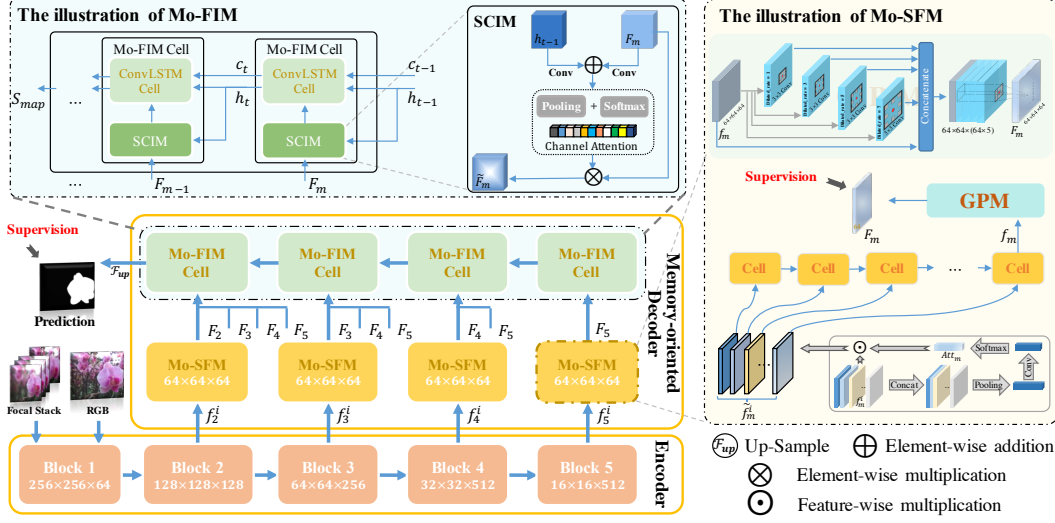

Figure 2: The overall architecture of our proposed network, which contains an encoder and a memory-oriented decoder.

shown in Fig. 2. In the encoder, RGB image is fed into a stream to generate raw RGB features while all focal slices are fed into another stream to generate light field features with abundant spacial information. For simplicity, we just illustrate one single encoder, which represents the two streams simultaneously. As suggested in [5], the Conv1_2 block (i.e., Block1) might be too shallow to make reliable prediction. We hereby perform our decoder on deeper layers (i.e., Block2-Block5). More specifically, given the RGB image $\mathcal{I}_0$ and the focal slices $\{\mathcal{I}_i\}_{i=1}^{12}$ with size $H \times W$, we denote the outputs of the last four blocks as $\{f_m^i, m = 2, 3, 4, 5\}_{i=0}^{12}$, where $i = 0$ represents features generated in the RGB stream, $i = 1 \cdots 12$ represents the indexes of focal slices and $m = 2, 3, 4, 5$ represents the last four convolution blocks.

## 4.2 The Memory-oriented Spatial Fusion Module (Mo-SFM)

With the raw RGB and light field features generated from the encoder, we aim at fusing all available information to address the challenging problem of light field SOD. A straightforward solution is to simply concatenate light field features produced by different focal slices. However, two drawbacks emerge in this approach. First, it ignores the relative contributions of different focal slices to the final results. Focal slices represent images focused at different depths in a scene as shown in Fig. 1. Intuitively, different focal slices have different weights regarding the salient objects. Second, direct concatenation operation seriously damages the spatial correlation of those focal slices. A more proper and effective fusion strategy should be considered. Hence, we propose a novel memory-oriented spatial fusion module (Mo-SFM) to address this problem. In this module, we introduce an attention mechanism shown in Fig. 2 to emphasize the useful features and suppress the unnecessary ones from focused and blurred information. This procedure can be defined as:

$$Att_m = \delta(\mathcal{W}_m * AvgPooling(\mathcal{D}[f_m^0; f_m^1; \cdots; f_m^{12}]) + b_m), \qquad (1)$$

$$\widetilde{f_m^i} = f_m^i \odot Att_m^i, i = 0, 1, \cdots, 12, \qquad (2)$$

where $\mathcal{D}[\cdot; \cdots; \cdot]$ means concatenation operation. $*$, $\mathcal{W}_m$ and $b_m$ represent convolution operator and convolution parameters in $m$-th layers. $AvgPooling(\cdot)$ means global average pooling operation and $\delta(\cdot)$ means softmax function. $Att_m \in \mathbb{R}^{1 \times 1 \times N}$ means the channel-wise attention map in $m$-th layers. $\odot$ denotes feature-wise multiplication.

Then those weighted light field features $\{\widetilde{f_m^i}\}_{i=0}^{12}$ are regarded as a sequence of inputs corresponding to the consecutive time steps. They are fed into a ConvLSTM [45] structure to gradually refine their

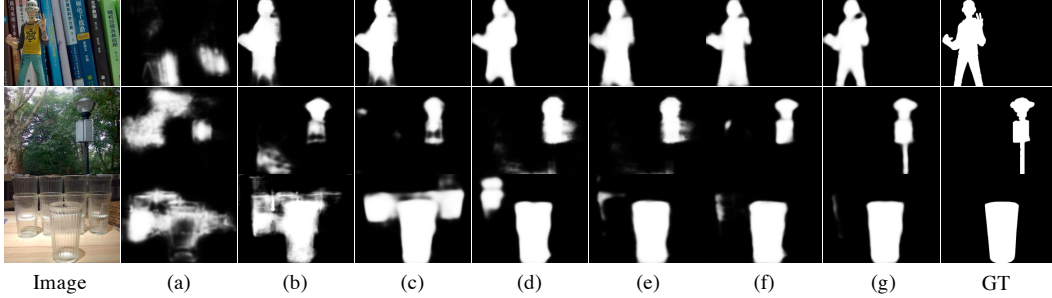

| Image | (a) | (b) | (c) | (d) | (e) | (f) | (g) | GT |

Figure 3: Visual comparisons in ablation studies. (a) means using RGB image only. (b) means using light field data (concatenation without weighting). (c) means concatenation with weighting. (d) means ConvLSTM fusion with weighting. (e) represents (d)+GPM (i.e., full Mo-SFM). (f) means our whole network without the SCIM. (g) means the final model.

spatial information for accurately identifying the salient objects. This procedure can be defined as:

$$
\begin{aligned}
i_t &= \sigma(\mathcal{W}_{xi} * \widetilde{f}_m^i + \mathcal{W}_{hi} * \mathcal{H}_{t-1} + \mathcal{W}_{ci} \circ \mathcal{C}_{t-1} + b_i), \\
f_t &= \sigma(\mathcal{W}_{xf} * \widetilde{f}_m^i + \mathcal{W}_{hf} * \mathcal{H}_{t-1} + \mathcal{W}_{cf} \circ \mathcal{C}_{t-1} + b_f), \\
\mathcal{C}_t &= f_t \circ \mathcal{C}_{t-1} + i_t \circ \tanh(\mathcal{W}_{xc} * \widetilde{f}_m^i + \mathcal{W}_{hc} * \mathcal{H}_{t-1} + b_c), \\
o_t &= \sigma(\mathcal{W}_{xo} * \widetilde{f}_m^i + \mathcal{W}_{ho} * \mathcal{H}_{t-1} + \mathcal{W}_{co} \circ \mathcal{C}_t + b_o), \\
\mathcal{H}_t &= o_t \circ \tanh(\mathcal{C}_t),
\end{aligned}
\tag{3}
$$

where $\circ$ denotes the Hadamard product and $\sigma(\cdot)$ is sigmoid function. A memory cell $\mathcal{C}_t$ stores the earlier information. All $\mathcal{W}_*$ and $b_*$ are model parameters to be learned. All the gates $i_t$, $f_t$, $o_t$, memory cell $\mathcal{C}_t$, and hidden state $\mathcal{H}_t$ are 3D tensors. In this way, after 13 steps, four fused light field features $\{f_2, f_3, f_4, f_5\}$ are effectively generated: $f_m = \mathcal{H}_{13}$. The unique property of the light field data makes it spontaneously suitable to use ConvLSTM for feature fusion. The ConvLSTM is also beneficial for making better use of the spatial correlation between multiple focal slices thanks to its powerful gate and memory mechanism. By now, our model enhances the average MAE performance by nearly 14.7% points on our proposed dataset and LFSD dataset (b $vs$ d in Tab. 1).

Furthermore, to capture global contextual information at different scales, we further extend a global perception module (GPM) on the top of $f_m$. The GPM can be defined as:

$$
F_m = Conv_{1 \times 1}(\mathcal{D}(f_m; \amalg_{r \in R^S}(Conv_d(f_m; \theta_m; r)))), m = 2, 3, 4, 5, \tag{4}
$$

where $\mathcal{D}[\ \cdot\ ;\ \cdots\ ;\ \cdot\ ]$ denotes concatenation operation. $\amalg_{r \in R^S}(\mathcal{OP})$ means operation, $\mathcal{OP}$ is performed several times using different dilation rates $r$ in $rates\_set$ (denoted as $R^S$) and all results are returned. $\theta_m$ is parameters to be learned in $m$-th layer. $\{F_m\}_{m=2}^5$ are the final fused light field features in multiple layers. At the end, we add several intermediate supervisions on $F_m$ in each layer to facilitate network convergence and encourage explicit fusion of those light field features.

### 4.3 The Memory-oriented Feature Integration Module (Mo-FIM)

Efficient integration of hierarchical deep features is significant for pixel-wise prediction tasks, e.g., salient object detection [60, 5], semantic segmentation [34]. We propose a new memory-oriented module, which from a novel perspective, utilizes the memory mechanism to effectively integrate multi-level light field features in a top-down manner. Specifically, as each channel of a feature map is considered as a 'feature detector' [53, 57], we design a scene context integration

Table 1: Quantitative results of the ablation analysis for our network. The meaning of indexes has been explained in the caption of Fig. 3.

|  |  | Ours | | LFSD | |
|---|---|---|---|---|---|
| indexes | Modules | $F_\beta \uparrow$ | $MAE \downarrow$ | $F_\beta \uparrow$ | $MAE \downarrow$ |
| (a) | RGB | 0.643 | 0.144 | 0.607 | 0.194 |
| (b) | LF(w/o weighting) | 0.805 | 0.074 | 0.781 | 0.121 |
| (c) | LF(with weighting) | 0.819 | 0.069 | 0.789 | 0.116 |
| (d) | +SFM(w/o GPM) | 0.821 | 0.062 | 0.797 | 0.105 |
| (e) | +SFM(with GPM) | 0.825 | 0.059 | 0.807 | 0.099 |
| (f) | +FIM(w/o SCIM) | 0.838 | 0.054 | 0.814 | 0.092 |
| (g) | +FIM(with SCIM) | **0.843** | **0.052** | **0.819** | **0.089** |

module (SCIM) shown in Fig. 2, which utilizes memory information from toper layers to learn a channel attention map and updates the current light field feature by focusing on important channels and suppressing unnecessary ones. Then, the ConvLSTM progressively integrates the high-level memories and the current elaborately refined input. That is to say, the high-level features with abundant semantic information are gradually summarized as memory and then being used to guide the selection of low-level spatial details for precise saliency prediction.

More specifically, in the SCIM shown in Fig. 2, $\mathcal{H}_{t-1}$ represents the previous scene understanding (i.e., hidden state of ConvLSTM in $t-1$ time step) and $F_m$ means the fused light field feature in $m^{th}$ layer. The SCIM can de defined as:

$$\widetilde{F}_m = \delta(AvgPooling(\mathcal{W}_1 * \mathcal{H}_{t-1} \oplus \mathcal{W}_2 * F_m)) \otimes F_m, \qquad (5)$$

where $\oplus$ and $\otimes$ denote element-wise addition and multiplication, respectively. Then the updated feature $\widetilde{F}_m$ is fed into a ConvLSTM cell to further summarize spatial information from the historical memory and current input $\widetilde{F}_m$. We use the output of Block5 as the initial state of ConvLSTM and SCIM, i.e., $H_0 = F_5$. After 4 steps (corresponding to $\widetilde{F}_5, \widetilde{F}_4, \widetilde{F}_3, \widetilde{F}_2$, respectively), the output of the ConvLSTM is followed by a transition convolutional layer and an up-sample operation to get the final saliency map $\mathcal{S}$. The calculation procedure is similar to Equ. 3 by replacing the inputs.

However, the top-down structure may cause high-level features diluted as they are transmitted to the lower layers. To address this problem, inspired by DenseNet [22], we link the features in low and high levels in the way shown in Fig. 2, to alleviate gradient vanishing and meanwhile encourage feature reuse. The final light field features to be used can be defined as: $F_m = \sum_{r=m}^{5} F_r$, $m$ is set to $2, 3, 4, 5$, successively. Besides, in order to guarantee each time step of the ConvLSTM can explicitly learn the most important information for accurately identifying salient objects, we add intermediate supervisions on all internal outputs of the ConvLSTM.

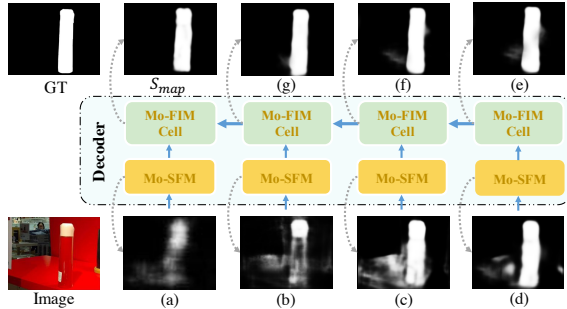

Figure 4: Visual results of the intermediate supervisions. In such a complex scene, our model can gradually optimize the saliency maps and produce a precise prediction.

Generally speaking, those intermediate supervisions can act as instruction to guide the SCIM and ConvLSTM to accurately filter the non-salient areas and retain salient areas. Intermediate results are illustrated in Fig. 4. Full details about codes will be made publicly available.

## 5 Experiments

### 5.1 Datasets

To evaluate the performance of our proposed network, we conduct experiments on our proposed dataset and the only two public light field saliency datasets: LFSD [29] and HFUT [59].

**Ours:** This dataset consists of 1462 light field samples. We randomly select 1000 samples for training and the remaining 462 samples for testing. More details can be found in Sec. 3.
**LFSD:** This dataset contains 100 light fields captured by Lytro camera. This dataset is proposed by Li *et al.*, in [29], which pioneered the use of light field for solving challenging problems in SOD.
**HFUT:** HFUT consists of 255 samples captured by Lytro camera. It is a challenging dataset, with the real-life scenarios at various distances, sensors noises, lighting conditions, and so on.

All samples in LFSD and HFUT are used for testing to evaluate the generalization abilities of saliency models. To avoid overfitting, we augment the training set by flipping, cropping and rotating.

### 5.2 Experiments Setup

**Evaluation Metrics.** We adopt five metrics for comprehensive evaluation, including Precision-Recall (PR) curve, F-measure [1], Mean Absolute Error (MAE), S-measure [12] and E-measure [13].

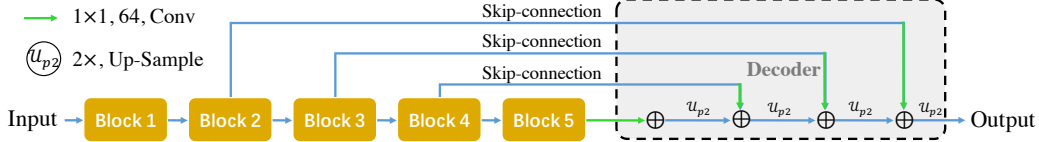

Figure 5: Illustration of the baseline network. Using RGB or light field data as input correspond to (a) and (b) in Fig. 3 and Tab. 1, respectively. In term of light field input, here, we use 'concatenation without weighting' strategy to fuse light field features from different focal slices in each Conv-Block. For fairness, the intermediate supervisions are same as our proposed network.

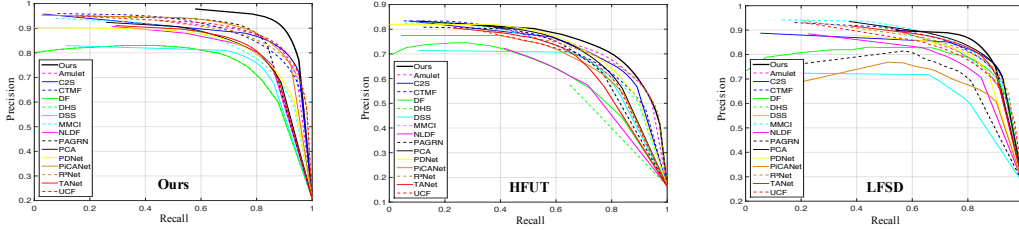

Figure 6: The PR curves of our proposed method and other CNNs-based methods. Obviously, ours is consistently outstanding over other approaches.

They are universally-agreed and standard for evaluating a SOD model and well explained in many literatures. Due to limited space, we will not show the detailed description.

**Implementation Details.** Our network is implemented on Pytorch framework and trained with a GTX 2080 Ti GPU. All training and test images are uniformly resized to $256 \times 256$. Our network is trained in an end-to-end manner, in which the momentum, weight decay and learning rate are set to 0.9, 0.0005, 1e-10, respectively. During the training phrase, we use softmax entropy loss, and the network is trained by standard SGD and converges after 40 epochs with batch size of 1. The two backbone networks of the RGB and focal stack streams are all initialized with corresponding pre-trained VGG-19 net [46]. Other parameters are initialized with Gaussian kernels.

## 5.3 Ablation Studies

**The Effectiveness of Light Field Data.** Tab. 1 (a) and (b) show the detection results of our baseline network illustrated in Fig. 5 with RGB data and with light field data, respectively. Numerical results measured by F-measure and MAE demonstrate that the network using light field data outperforms the one only using RGB data. Fig. 3 (a) and (b) show the visual comparisons of two aforementioned networks, respectively. This also indicates that light field data improve prediction performance under challenging circumstances. Moreover, we conduct an experiment by repeating the RGB input-frame 12 times, in such a way that the model architecture is identical to the 4D version but the input data is only 2D. The quantitative results in term of F-measure and MAE are 0.819 / 0.089 (focal slices) and 0.740 / 0.140 (RGB) respectively. This further confirms the effectiveness of the focusness information and our spatial fusion module.

**The Effectiveness of Mo-SFM.** To give evidence for the effectiveness of the Mo-SFM, we compare the baseline network with it adding the Mo-SFM. Significant improvement can be visually observed between them shown in Fig. 3 (b) and (e). Numerically, our Mo-SFM reduces the MAE performances by nearly 19.2% on two datasets. To conduct further investigation, we provide internal inspection on the Mo-SFM. The gradual improvements, as we add our feature weighting mechanism, ConvLSTM integrator and the GPM into the Mo-SFM shown in Fig. 3 (c), (d) and (e), are consistent with our assertion that different contributions and spatial correlation of different focal slices are beneficial to SOD. Also, GPM is proved to be able to adaptively detect objects of different scales. Quantitative results in Tab. 1 also numerically show the accumulative accuracy gains from the three components.

**The Effectiveness of Mo-FIM.** The Mo-FIM is proposed for higher cognitive processing. Fig. 3 (g) visually shows the influence of the addition of the Mo-FIM. We observe that considerable gains (reduce the MAE by 11.8% and 10.1% shown in Tab. 1) are achieved. This result is logical since high-level features are gradually summarized as memory and then being used to guide the selection of low-level spatial details by using the Mo-FIM. Results in Fig. 3 show that removing the SCIM from the Mo-FIM may lead to false positives. This suggests that the SCIM effectively updates the original input according to memory-oriented scene understanding and may greatly bias the results.

Table 2: Quantitative comparisons on the light field datasets. The best three results are shown in **boldface**, red, and green fonts respectively. $^*$ means non-deep-learning. - means no available results.

| Types | Methods | Years | Ours | | | | HFUT [59] | | | | LFSD [29] | | | |
|---|---|---|---|---|---|---|---|---|---|---|---|---|---|---|
| | | | $E_s\uparrow$ | $S_\alpha\uparrow$ | $F_\beta\uparrow$ | MAE↓ | $E_s\uparrow$ | $S_\alpha\uparrow$ | $F_\beta\uparrow$ | MAE↓ | $E_s\uparrow$ | $S_\alpha\uparrow$ | $F_\beta\uparrow$ | MAE↓ |
| 4D | Ours | - | **0.923** | **0.887** | **0.843** | **0.052** | **0.785** | **0.742** | **0.627** | **0.095** | **0.886** | **0.830** | **0.819** | **0.089** |
| | LFS$^*$ [29] | TPAMI'17 | 0.728 | 0.563 | 0.484 | 0.240 | 0.650 | 0.559 | 0.416 | 0.222 | 0.771 | 0.680 | 0.740 | 0.208 |
| | MCA$^*$ [59] | TOMM'17 | - | - | - | - | 0.714 | 0.652 | 0.558 | 0.139 | 0.841 | 0.749 | 0.815 | 0.150 |
| | WSC$^*$ [56] | CVPR'15 | - | - | - | - | - | - | - | - | 0.794 | 0.706 | 0.706 | 0.156 |
| | DILF$^*$ [58] | IJCAI'15 | 0.805 | 0.705 | 0.641 | 0.168 | 0.701 | 0.669 | 0.529 | 0.148 | 0.810 | 0.755 | 0.728 | 0.168 |
| 3D | TANet [6] | TIP'19 | 0.861 | 0.803 | 0.771 | 0.096 | 0.761 | 0.711 | 0.605 | 0.111 | 0.849 | 0.803 | 0.804 | 0.112 |
| | MMCI [7] | PR'19 | 0.853 | 0.785 | 0.750 | 0.116 | 0.748 | 0.711 | 0.608 | 0.116 | 0.848 | 0.799 | 0.796 | 0.128 |
| | PCA [5] | CVPR'18 | 0.857 | 0.800 | 0.762 | 0.100 | 0.757 | 0.730 | 0.619 | 0.104 | 0.846 | 0.807 | 0.801 | 0.112 |
| | PDNet [65] | arXiv'18 | 0.864 | 0.803 | 0.763 | 0.111 | 0.758 | 0.741 | 0.608 | 0.112 | 0.849 | 0.786 | 0.780 | 0.116 |
| | CTMF [17] | TCyb'17 | 0.881 | 0.823 | 0.790 | 0.100 | 0.747 | 0.723 | 0.596 | 0.119 | 0.856 | 0.801 | 0.791 | 0.119 |
| | DF [42] | TIP'17 | 0.838 | 0.716 | 0.733 | 0.151 | 0.701 | 0.641 | 0.531 | 0.156 | 0.816 | 0.751 | 0.756 | 0.162 |
| | CDCP$^*$ [66] | ICCVW'17 | 0.795 | 0.690 | 0.639 | 0.159 | 0.696 | 0.653 | 0.528 | 0.159 | 0.739 | 0.659 | 0.642 | 0.201 |
| | ACSD$^*$ [25] | ICIP'15 | 0.629 | 0.385 | 0.151 | 0.321 | 0.665 | 0.559 | 0.421 | 0.201 | 0.803 | 0.731 | 0.764 | 0.185 |
| | NLPR$^*$ [39] | ECCV'14 | 0.768 | 0.564 | 0.659 | 0.177 | 0.706 | 0.579 | 0.567 | 0.148 | 0.744 | 0.553 | 0.712 | 0.216 |
| 2D | PiCANet [36] | CVPR'18 | 0.892 | 0.829 | 0.821 | 0.083 | 0.762 | 0.719 | 0.600 | 0.115 | 0.780 | 0.729 | 0.671 | 0.158 |
| | PAGRN [62] | CVPR'18 | 0.878 | 0.822 | 0.828 | 0.084 | 0.758 | 0.704 | 0.619 | 0.116 | 0.805 | 0.727 | 0.725 | 0.147 |
| | C2S [31] | ECCV'18 | 0.874 | 0.844 | 0.791 | 0.084 | 0.762 | 0.736 | 0.618 | 0.112 | 0.820 | 0.806 | 0.749 | 0.113 |
| | R$^3$Net [11] | IJCAI'18 | 0.833 | 0.819 | 0.783 | 0.113 | 0.697 | 0.720 | 0.606 | 0.151 | 0.838 | 0.789 | 0.781 | 0.128 |
| | Amulet [60] | ICCV'17 | 0.882 | 0.847 | 0.805 | 0.083 | 0.737 | 0.739 | 0.615 | 0.118 | 0.821 | 0.773 | 0.757 | 0.135 |
| | UCF [61] | ICCV'17 | 0.850 | 0.837 | 0.769 | 0.107 | 0.729 | 0.736 | 0.596 | 0.144 | 0.776 | 0.762 | 0.710 | 0.169 |
| | NLDF [37] | CVPR'17 | 0.862 | 0.786 | 0.778 | 0.103 | 0.761 | 0.685 | 0.583 | 0.107 | 0.810 | 0.745 | 0.748 | 0.138 |
| | DSS [21] | CVPR'17 | 0.827 | 0.764 | 0.728 | 0.128 | 0.759 | 0.699 | 0.606 | 0.138 | 0.749 | 0.677 | 0.644 | 0.190 |
| | DHS [35] | CVPR'16 | 0.872 | 0.841 | 0.801 | 0.090 | 0.720 | 0.642 | 0.542 | 0.129 | 0.836 | 0.770 | 0.761 | 0.133 |
| | MST$^*$ [49] | CVPR'16 | 0.785 | 0.686 | 0.629 | 0.157 | 0.693 | 0.641 | 0.529 | 0.156 | 0.754 | 0.659 | 0.631 | 0.191 |
| | BSCA$^*$ [41] | CVPR'15 | 0.811 | 0.720 | 0.690 | 0.180 | 0.693 | 0.651 | 0.530 | 0.193 | 0.777 | 0.718 | 0.688 | 0.203 |
| | DSR$^*$ [30] | ICCV'13 | 0.799 | 0.678 | 0.645 | 0.164 | 0.695 | 0.655 | 0.518 | 0.153 | 0.736 | 0.633 | 0.631 | 0.208 |

**The Limitations of Our Approach.** In this paper, we present a deep-learning-based light field saliency detection method for deeply exploring and comprehensively exploiting internal correlation of focal slices. We demonstrate the success of our method by achieving the state-of-the-art on three datasets. We see this work as opening two potential directions for future study. The first is building a big and versatile dataset for training and validating different models. We present one dataset-training our model and testing other 2D, 3D and 4D models-but one could also be bigger for improving generalization ability of all the models training on it. The other direction is developing a more computation-efficient and memory-efficient method as the focal stack is employed in the training process. We present the first deep-learning-based method for light field saliency detection, but there are other lightweight models that could potentially benefit from the light field data.

## 5.4   Comparisons with State-of-the-arts

We compare results from our method and other 25 2D, 3D and 4D ones, containing both deep-learning-based methods and non-deep learning ones(remarked with $^*$). There are 4 4D light field methods: LFS$^*$ [29], MCA$^*$ [59], WSC$^*$ [56], DILF$^*$ [58]; 9 3D RGB-D methods: TANet [6], MMCI [7], PCA [5], PDNet [65], CTMF [17], DF [42], CDCP$^*$ [66], ACSD$^*$ [25], NLPR$^*$ [39]; and 12 top-ranking RGB methods: PiCANet [36], PAGR [62], C2S [31], R$^3$Net [11], Amulet [60], UCF [61], NLDF [37], DSS [21], DHS [35], MST$^*$ [49], BSCA$^*$ [41], DSR$^*$ [30]. For fair comparisons, the results from competing methods are generated by authorized codes or directly provided by authors.

**Quantitative Evaluation.** Quantitative results are shown in Tab. 2. The proposed model consistently achieves the highest scores on all datasets across four evaluation metrics. An important observation should be noted: compared to the latest CNNs-based RGB SOD methods with large-quantity training sets, our method also achieves significant advantages with a relatively small training set. This indicates that light field data are significant and promising. Fig. 6 shows that the PR curves of our method outperform those top-ranking approaches.

**Qualitative Evaluation.** Fig. 7 shows some selected representative samples of results comparing our method with those of the current state-of-the-art methods. Our method is able to handle a wide rage of challenging scenes, including shown in Fig. 7, small objects ($1^{st}$ row), similar foreground and background ($2^{nd}$, $4^{th}$ and $9^{th}$ rows), clutter background ($3^{rd}$-$5^{th}$ and $8^{th}$ rows), and other difficult scenes ($6^{th}$ and $7^{th}$ rows). In those complex cases, we can see that our predicted results can be positively influenced by the light field data and our proposed network where the light field features from different focal slices are effectively fused and the multi-level global semantic information and local detail cues are sufficiently integrated.

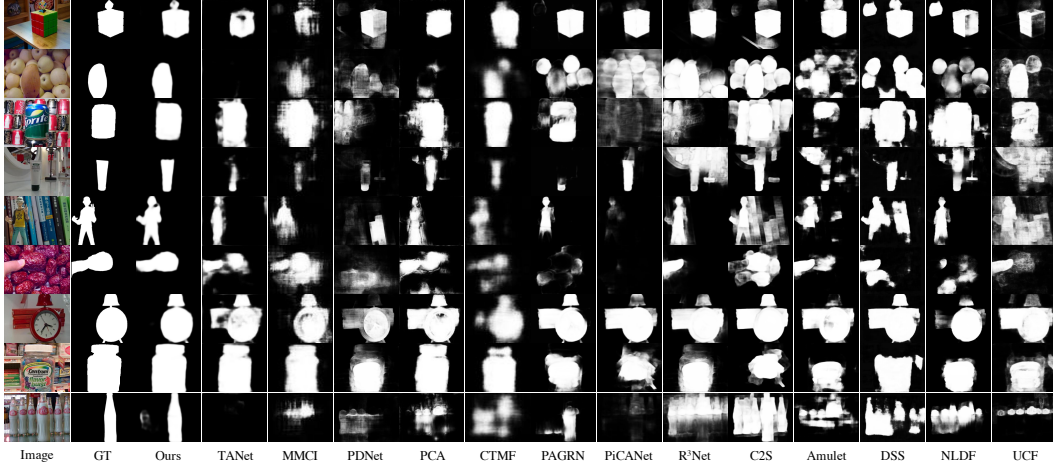

Image GT Ours TANet MMCI PDNet PCA CTMF PAGRN PiCANet R³Net C2S Amulet DSS NLDF UCF

Figure 7: Visual comparisons of our method with top-ranking CNNs-based methods in some challenging cases. Obviously, our model can generate precise salient results even in those complex scenes, which indicates that our method takes full advantages of light fields for accurate saliency prediction.

## 6 Conclusion

In this paper, we develop a novel memory-oriented decoder tailored for light field saliency detection. Our Mo-SFM resembles the memory mechanism of how human fuse information and effectively excavates the various contributions and spatial correlations of different focal slices. The Mo-FIM also sufficiently integrates multi-level features by leveraging high-level memory to guide low-level selection. Additionally, we introduce a large-scale light field saliency dataset to pave the way for future studies. Experiments show that our method achieves superior performance over 25 methods including 2D, 3D and 4D ones, especially in complex scenarios.

## Acknowledgements

This work was supported by the National Natural Science Foundation of China (61605022 and 61976035) and the Fundamental Research Funds for the Central Universities (DUT19JC58). The authors are grateful to the reviewers for their suggestions in improving the quality of the paper.

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
