[Reviews · NeurIPS 2019]

Reviewer 1



- - - - - - - - - post-rebuttal - - - - - - - - - - After reading the rebuttal and the other reviews, I am happy to recommend acceptance of this paper. My concerns were answered in the rebuttal and I do not see any major concerns in the other reviews. The provided explanations and results should be integrated into the final version. - - - - - - - - - original review - - - - - - - - - - W1 - Evaluation W1.1 The 2D model has much less capacity than the 3D/4D model. Thus, one could evaluate how much the LSTM module contributes by repeating the RGB input-frame 12 times. This way the model architecture is identical to the 4D version but the input data is only 2D. A direct comparison would then be possible. W1.2 How are the results for other (learning based) methods in Tab 2 obtained on the dataset that is introduced with this submission? Are they retrained and then evaluated or are the pretrained models evaluated on the new data? The later case would make the evaluation not directly comparable since different data was used. W2 - LSTM One of the main advantages of sequence models is that they can deal with variable lengths. Here, every “sequence” is of fixed length and the order of focus planes is not really crucial for the task. So the question arises if there is a simpler model that can deal with the presented data. For example a temporal convolution approach could also be used to fuse feature map across time steps and would reduce the complexity of the model. An order agnostic operation such as max/average-pooling would make the method invariant to the order of input images. W3 - Clarity W3.1 Table 1 is very tedious to read since one has to refer to the caption of Fig 3 to understand the row labels. A better solution would be including a short-hand notation for the ablated components that are written in the appropriate column (also Fig 3 could benefit from this). W3.2 The notation is slightly difficult to follow since there is often double and triple indexing, variables, indices and functions using words or multiple letters (Att, Conv_dilated, rates_set). I.e. operators should be typeset appropriately to make a proper visual distinction to variables and their multiplication.

Reviewer 2



Originality: my feeling it that the proposed neural network has some novelty, but the authors did not position clearly the paper with respect to the related work. The proposed architecture includes many components (such as the Memory-oriented Spatial Fusion Module, The Memory-oriented Integration Module ) and it’s not clear if they are original or not. I’m not an expert in the field, but I see some similarity with [29]. The originality of each component, and of the overall architecture, should be discussed and compared with similar ones. Quality: yes, I think the paper is technically sound, the method justified enough. The proposed method is supported by the results. I would expect the authors to highlight the limitations of their approach, which is not the case. Clarity: the paper reads well, even if it’s very dense and sometimes hard to follow (due to the density). But on overall, I think all the elements given in the papers can be understood, and they cover the proposed neural network well. Significance: the results are very good on the task of SOD for Light Field images, I have no doubts about it. But what about other areas? Does it also give good results on hyper spectral images? Competing papers have shown their approaches can be successfully used in several domains, in this paper only one domain is used, which limits the significance of the paper.

Reviewer 3



Originality: Moderate. Previous methods have demonstrated the light field image can improve saliency detection. So the main novelty in this paper might be using a deep network to solve this problem. Using the ConvLSTM to fuse features from different focal lengths and different levels of VGG is interesting, but not a significant innovation. Quality: High. The demonstration of the method is clear and related works seem to be complete. Also, there ae comprehensive experiments for ablation study and to compare with previous state-of-the-art. The experiments results are convincing and thorough. Clarity: The paper is easy to read and well organized. Figure 2 demonstrates the network structure clearly and concisely. All necessary details and explanations have been included in the submission. Significance: Given better performance compared with previous works, as well as indicated release of the dataset and code, his paper may be of interest to other researchers working on similar topics. Minor issues: On line 185, is t-1 equal to 13? The intuition of using H_t-1 in SCIM is not clear.

[Author Response · NeurIPS 2019]

We greatly appreciate the time and effort each of the reviewers have dedicated to providing insightful feedback on ways to strengthen our paper. Thus, it is with great pleasure that we make a point-by-point response to all comments.

**To Reviewer1: W1.1**: Thank you for your suggestion. We agree that a direct comparison between 12 RGB input-frames and 12 focal slices could highlight our spatial fusion module. 12 RGB input-frames containing the same information do not enable the SFM to learn the spatial correlation as well as the focal slices do. This is consistent with the results shown in the table below. We compare the results using two different inputs in terms of S-measure, F-measure and MAE. The results confirm the effectiveness of focusness information and our spatial fusion module. **W1.2**: We totally agree with you on that point. In our case, the 2D dataset (DUTS/MSRA10K) is 10 times larger than the proposed dataset. It is well understood that training dataset size is the most important factor determining the generalization ability regardless of model complexity. Based on this observation, we think it would be fairer to train different models on the same training dataset. Thus, it could provide an unbiased estimation for them tested on other datasets. The following table shows the results, tested on another dataset (LSFD), from the models we have retrained on our dataset. It is shown that the retrained models on our dataset do not perform as well as they are pre-trained on the much larger 2D dataset. This is the reason we choose more generalized models to compare with ours.

**W2:** Thank you for providing us with the constructive comments. In this paper, we adopted a simple model in Table 1(c) (concatenation followed by a $1 \times 1$ convolution layer) to replace the Mo-SFM for feature fusion. Experimental results indicates that the proposed Mo-SFM has superiority in learning spatial correlation of different features and performs better.

**W3:** We are sorry that the notation and display of results made the paper less readable than intended. We will carefully improve them in the final version according to your suggestion.

| Score on LFSD[29] | Ours | | PDNet[65] | | DHS[35] | | NLDF[37] | | UCF[61] | | Amulet[60] | | R$^3$Net[11] | |
| --- | --- | --- | --- | --- | --- | --- | --- | --- | --- | --- | --- | --- | --- | --- |
| | focal slices | RGB*12 | w/o | with | w/o | with | w/o | with | w/o | with | w/o | with | w/o | with |
| $S_{measure} \uparrow$ | 0.830 | 0.757 | 0.786 | 0.746 | 0.770 | 0.741 | 0.745 | 0.731 | 0.762 | 0.748 | 0.773 | 0.757 | 0.789 | 0.769 |
| $F_{measure} \uparrow$ | 0.819 | 0.740 | 0.780 | 0.734 | 0.761 | 0.732 | 0.748 | 0.720 | 0.710 | 0.702 | 0.757 | 0.738 | 0.781 | 0.759 |
| $MAE \downarrow$ | 0.089 | 0.140 | 0.116 | 0.136 | 0.133 | 0.142 | 0.138 | 0.137 | 0.169 | 0.174 | 0.135 | 0.147 | 0.128 | 0.137 |

∗ 'w/o' means no-retrain results, and 'with' means retrain results.

**To Reviewer2: About the originality:** [29] as the pioneering work is a traditional method in which various hand-crafted features and prior knowledges are used for light field saliency detection. However, deep-learning-based light field methods have been missing from contemporary studies in saliency detection. We first introduce the CNN framework for light field SOD and provide the largest light field dataset. The overall architecture of our paper differs substantially from [29] and other light field saliency detection methods. As to the proposed modules, the Mo-SFM resembles the memory mechanism to fuse information by emphasizing useful features and learning the spatial correlation between those light field features. This is the first attempt in light field SOD. Furthermore, different from other integration mechanisms for SOD, such as skip-connections [35,36], short-connections [21], and residual connections [11], our Mo-FIM is designed in a way high-level information is summarized as memory to guide low-level feature selection. To our best knowledge, this feature integration has never been explored in previous studies.

**About the limitations:** We appreciate the opportunity to include additional explanation about the limitation of our paper. First, our dataset is not big enough compared with other 2D datasets. This may limit the generalization ability of models training on it. Second, the use of the focal stack in the training process does require higher-end hardware devices or is a bit more time-consuming. We are currently working on these two limitations.

**About the use of domains:** We believe that there is a large space to explore our method in different fields, because Mo-SFM can be used to effectively fuse sequent features, and Mo-FIM can be applied to integrate multi-layer features for dense prediction tasks. However, deep learning based light field SOD is still in the initial stage. In the experiment, our main focus in this work lies on demonstrating the rationality and feasibility of the method. For the hyper spectral images, we suppose that our model can not be directly used on hyper spectral images due to different data types.

**To Reviewer3:** Thank you for the positive and constructive comments. We concur with you that a detailed description of our dataset would be important for NeurIPS readers and further strengthen the paper. To build this dataset, we first capture 3894 images under different lighting conditions in various environments around our daily life, e.g., parks, campus, streets and supermarkets and so on. Then we screen out blurred images or low-quality images. Acquisition of our ground truths follows annotation principle in the widely used saliency datasets, e.g., LFSD and DUTS. Because determining salient objects is considered highly subjective, three annotators are required to determine the salient objects in order to achieve annotation consensus. A free image annotation tool-GIMP is used manually to annotate the salient objects in a pixel-wise manner. Then 1462 light fields are selected as the final dataset, consisting of 900 indoor and 562 outdoor scenes. Moreover, it contains challenging scenes, e.g., multiple objects (95), transparent objects (28), low-contrast or similar foreground and background (108), low-intensity environment (9), and clutter background (31).

**For the minor issues,** we are sorry that they are not clearly described here. To be specific, we use the output of $Block5$ as the initial hidden state of ConvLSTM, i.e., $h_0 = F_5$. In the multi-level feature integration stage, $h_{t-1}$ is considered as historical memory to learn a channel attention and update current light field features in SCIM. We will clarify the minor issues and incorporate a more clear description of our dataset in the final version.

Again, thank you all for giving us the opportunity to strengthen our paper with your valuable comments and queries. We will make the code and dataset publicly available. Hopefully this would maximize the contribution to the community.

[Meta-Review · NeurIPS 2019]

The paper provided an interesting method for salient object segmentation in light field images backed by solid empirical evaluation. All reviewers are in favor of acceptance.